# Double Pessimism is Provably Efficient for Distributionally Robust Offline Reinforcement Learning: Generic Algorithm and Robust Partial Coverage

**Jose Blanchet**[1*]    **Miao Lu**[1*]    **Tong Zhang**[2*]    **Han Zhong**[3*]

[1] Department of Management Science and Engineering, Stanford University
[2] Department of Mathematics, The Hong Kong University of Science and Technology
[3] Center for Data Science, Peking University

## Abstract

We study distributionally robust offline reinforcement learning (RL), which seeks to find an optimal robust policy purely from an offline dataset that can perform well in perturbed environments. We propose a generic algorithm framework Doubly Pessimistic Model-based Policy Optimization ($P^2MPO$) for robust offline RL, which features a novel combination of a flexible model estimation subroutine and a doubly pessimistic policy optimization step. Here the *double pessimism* principle is crucial to overcome the distribution shift incurred by i) the mismatch between behavior policy and the family of target policies; and ii) the perturbation of the nominal model. Under certain accuracy assumptions on the model estimation subroutine, we show that $P^2MPO$ is provably sample-efficient with *robust partial coverage data*, which means that the offline dataset has good coverage of the distributions induced by the optimal robust policy and perturbed models around the nominal model. By tailoring specific model estimation subroutines for concrete examples including tabular Robust Markov Decision Process (RMDP), factored RMDP, and RMDP with kernel and neural function approximations, we show that $P^2MPO$ enjoys a $\widetilde{\mathcal{O}}(n^{-1/2})$ convergence rate, where $n$ is the number of trajectories in the offline dataset. Notably, these models, except for the tabular case, are first identified and proven tractable by this paper. To the best of our knowledge, we first propose a general learning principle — double pessimism — for robust offline RL and show that it is provably efficient in the context of general function approximations.

## 1 Introduction

Reinforcement learning (RL) [52] aims to learn an optimal policy that maximizes the cumulative rewards received in an unknown environment. Typically, deep RL algorithms learn a policy in an online trial-and-error fashion using millions to billions of data. However, data collection could be costly and risky in some practical applications such as healthcare [56] and autonomous driving [38]. To tackle this challenge, offline RL (also known as batch RL) [21, 22] learns a near-optimal policy based on a dataset collected a priori without further interactions with the environment. Although there has been great progress in offline RL [72, 20, 16, 54, 62, 6], these works implicitly require that the offline dataset is generated by the real-world environment, which may fail in practice. Taking robotics [18, 37] as an example, the experimenter trains agents in a simulated physical environment and then deploy them in real-world environments. Since the experimenter does not have access to the true physical environment, there is a mismatch between the simulated environment used to generate the offline dataset and the real-world environment used to deploy the agents. Such a mismatch is

---

*Alphabetical order. Email to `miaolu@stanford.edu`

37th Conference on Neural Information Processing Systems (NeurIPS 2023).

commonly referred to as the *sim-to-real gap* [42, 76]. Since the optimal policy is sensitive to the model [31, 9], the potential sim-to-real gap may lead to the poor performance of RL algorithms.

A promising solution to remedy this issue is robust RL [13, 9, 32] – training a robust policy that performs well in a bad or even adversarial environment. A line of work on deep robust RL [43, 44, 41, 30, 53, 75, 19] demonstrates the superiority of the trained robust policy in real world environments. Furthermore, the recent work of Hu et al. [11] theoretically proves that the ideal robust policy can attain near optimality in dealing with problems with sim-to-real gap, but this work does not suggest how to learn a robust policy from a theoretical perspective. In order to understand robust RL from the theoretical side, robust Markov decision process (RMDP) [13, 9] has been proposed, and many recent works [77, 47, 29] design sample-efficient learning algorithms for robust offline RL. These works mainly focus on the tabular case, which is not capable of tackling large state spaces. Meanwhile, in the non-robust setting, a line of works [16, 54, 62, 73, 45] show that "pessimism" is the general learning principle for designing algorithms that can overcome the distributional shift problem faced by offline RL. In particular, in the context of function approximation, Xie et al. [62] and Uehara and Sun [54] leverage the pessimism principle and propose generic algorithms in the model-free and model-based fashion, respectively. Hence, it is natural to ask the following questions:

**Q1:** What is the general learning principle for robust offline RL?
**Q2:** Based on this learning principle, can we design a generic algorithm for robust offline RL in the context of function approximation?

To answer these two questions, we need to tackle the following two intertwined challenges: (i) distributional shift, that is, the mismatch between offline data distribution and the distribution induced by the optimal robust policy. In robust offline RL, the distributional shift has two sources – behavior policy and perturbed model, where the latter is the unique challenge not presented in non-robust RL; and (ii) function approximation. Existing works mainly focus on the tabular case, and it remains elusive how to add reasonable structure conditions to make RMDPs with large state spaces tractable. Despite these challenges, we answer the aforementioned two questions affirmatively.

**Contributions.** We study robust offline RL in a general framework, which not only includes existing known tractable $\mathcal{S} \times \mathcal{A}$-rectangular tabular RMDPs, but also subsumes several newly proposed models (e.g., $\mathcal{S} \times \mathcal{A}$-rectangular factored RMDPs, $\mathcal{S} \times \mathcal{A}$-rectangular kernel RMDPs, and $\mathcal{S} \times \mathcal{A}$-rectangular neural RMDPs) as special cases. Under this general framework, we propose a generic model-based algorithm, dubbed as Doubly Pessimistic Model-based Policy Optimization (P$^2$MPO), which consists of a model estimation subroutine and a policy optimization step based on *doubly pessimistic* value estimators. We note that the model estimation subroutine can be flexibly chosen according to structural conditions of specific RMDP examples. Meanwhile, the adoption of doubly pessimistic value estimators in the face of model estimation uncertainty and environment uncertainty plays a key role in overcoming the distributional shift problem in robust offline RL.

From the theoretical perspective, we characterize the optimality of P$^2$MPO with partial coverage. In particular, we show that the suboptimality gap of P$^2$MPO is upper bounded by the model estimation error (see Condition 3.2) and the robust partial coverage coefficient (see Assumption 3.3). For concrete examples of RMDPs, by customizing specific model estimation mechanisms and plugging them into P$^2$MPO, we show that P$^2$MPO enjoys a $n^{-1/2}$ convergence rate with robust partial coverage data, where $n$ is the number of trajectories in the offline dataset. In summary, we identify a general learning principle — *double pessimism* — for robust offline RL. Based on this principle, we can perform sample-efficient robust offline RL with robust partial coverage data via general function approximation. See Table 1 for a summary of our results and a comparison with mostly related works.

## 1.1 Related Works

**Robust reinforcement learning in robust Markov decision processes.** Robust RL is usually modeled as a robust MDP (RMDP) [13, 9], and its planning has been well studied [13, 9, 65, 60, 57]. Recently, robust RL in RMDPs has attracted considerable attention, and a growing body of works studies this problem in the generative model [68, 39, 49, 58, 69, 66, 7], online setting [59, 3, 8], and offline setting [77, 40, 47, 29]. Our work focuses on robust offline RL, and we provide a more in-depth comparison with Zhou et al. [77], Shi and Chi [47], Ma et al. [29] as follows. Under the full coverage condition (a uniformly lower bounded data distribution), Zhou et al. [77] provide the first sample-efficient algorithm for $\mathcal{S} \times \mathcal{A}$-rectangular tabular RMDPs. After, Shi and Chi [47]

Table 1: A comparison with closely related works on robust offline RL. ✓ means the work can tackle this model with robust partial coverage data, ✓! means the work requires full coverage data to solve the model, and ✗ means the work cannot tackle the model. Lightblue color denotes the models that are first proposed and proved tractable in this work.

| | Zhou et al. [77] | Shi and Chi [47] | Ma et al. [29] | This Work |
|---|---|---|---|---|
| $\mathcal{S} \times \mathcal{A}$-rectangular tabular RMDP | ✓! | ✓ | ✗ | ✓ |
| $d$-rectangular linear RMDP | ✗ | ✗ | ✓ | ✓ |
| $\mathcal{S} \times \mathcal{A}$-rectangular factored RMDP | ✗ | ✗ | ✗ | ✓ |
| $\mathcal{S} \times \mathcal{A}$-rectangular kernel RMDP | ✗ | ✗ | ✗ | ✓ |
| $\mathcal{S} \times \mathcal{A}$-rectangular neural RMDP | ✗ | ✗ | ✗ | ✓ |

leverage the pessimism principle and design a sample-efficient offline algorithm that only requires robust partial coverage data for $\mathcal{S} \times \mathcal{A}$-rectangular tabular RMDPs. Ma et al. [29] propose a new $d$-rectangular RMDP and develop a pessimistic style algorithm that can find a near-optimal robust policy with partial coverage data. In comparison, we provide a generic algorithm that can not only solve the models in Zhou et al. [77], Shi and Chi [47], Ma et al. [29], but can also tackle various newly proposed RMDP models such as $\mathcal{S} \times \mathcal{A}$-rectangular factored RMDP, $\mathcal{S} \times \mathcal{A}$-rectangular kernel RMDP, and $\mathcal{S} \times \mathcal{A}$-rectangular neural RMDP. See Table 1 for a summary. Moreover, we propose a new pessimistic type learning principle "double pessimism" for robust offline RL. Although Shi et al. [48] and Ma et al. [29] adopt the similar algorithmic idea in tabular or linear settings, neither of them have identified a general learning principle for robust offline RL in the regime of large state space.

**Non-robust offline RL and pessimism principle.** The line of works on offline RL aims to design efficient learning algorithms that find an optimal policy given an offline dataset collected a priori. Prior works [33, 2, 5] typically require a dataset of full coverage, which assumes that the offline data have good coverage of all state-action pairs. In order to avoid such a strong coverage condition on data, the *pessimism* principle – being conservative in policy or value estimation of those state-action pairs that are not sufficiently covered by data – has been proposed. Based on this principle, a long line of works [see e.g., 16, 54, 62, 63, 45, 73, 71, 64, 48, 24, 26, 74, 28, 46] propose algorithms that can learn the optimal policy only with the *partial coverage data*. The partial coverage data only requires to cover the state-action pairs visited by the optimal policy. Among these works, our work is mostly related to the work of Uehara and Sun [54], which proposes a generic model-based algorithm for non-robust offline RL. Our algorithm for robust offline RL is also in a model-based fashion, and our study covers some models such as $\mathcal{S} \times \mathcal{A}$-rectangular kernel and neural RMDPs whose non-robust counterparts are not studied by Uehara and Sun [54]. More importantly, our algorithm is based on a newly proposed *double pessimism* principle, which is tailored for robust offline RL and is in parallel with the pessimism principle used in non-robust offline RL. Also, we show that the performance of our proposed algorithm depends on the notion of *robust partial coverage coefficient*, which is also different from the notions of partial coverage coefficient in previous non-robust offline RL works [16, 62, 54].

## 1.2 Notations

For any set $A$, we use $2^A$ to denote the collection of all the subsets of $A$. For any measurable space $\mathcal{X}$, we use $\Delta(\mathcal{X})$ to denote the collection of probability measures over $\mathcal{X}$. For any integer $n$, we use $[n]$ to denote the set $\{1, \cdots, n\}$. Throughout the paper, we use $D(\cdot\|\cdot)$ to denote a (pseudo-)distance between two probability measures (or densities). In specific, we define the KL-divergence $D_{\mathrm{KL}}(p\|q)$ between two probability densities $p$ and $q$ over $\mathcal{X}$ as

$$D_{\mathrm{KL}}(p\|q) = \int_{\mathcal{X}} p(x) \log\left(\frac{p(x)}{q(x)}\right) \mathrm{d}x,$$

and we define the TV-distance $D_{\mathrm{TV}}(p\|q)$ between two probability densities $p$ and $q$ over $\mathcal{X}$ as

$$D_{\mathrm{TV}}(p\|q) = \frac{1}{2} \int_{\mathcal{X}} |q(x) - p(x)| \, \mathrm{d}x.$$

Given a function class $\mathcal{F}$ equipped with some norm $\|\cdot\|_{\mathcal{F}}$, we denote by $\mathcal{N}_{[]}(\epsilon, \mathcal{F}, \|\cdot\|_{\mathcal{F}})$ the $\epsilon$-bracket number of $\mathcal{F}$, and $\mathcal{N}(\epsilon, \mathcal{F}, \|\cdot\|_{\mathcal{F}})$ the $\epsilon$-covering number of $\mathcal{F}$.

## 2 Preliminaries

### 2.1 A Unified Framework of Robust Markov Decision Processes

We introduce a unified framework for studying episodic robust Markov decision processes (RMDP), which we denote as a tuple $(\mathcal{S}, \mathcal{A}, H, P^\star, R, \mathcal{P}_{\mathrm{M}}, \mathbf{\Phi})$. The set $\mathcal{S}$ is the state space with possibly infinite cardinality, $\mathcal{A}$ is the action space with finite cardinality. The integer $H$ is the length of each episode. The set $P^\star = \{P_h^\star\}_{h=1}^H$ is the collection of transition kernels where each $P_h^\star : \mathcal{S} \times \mathcal{A} \mapsto \Delta(\mathcal{S})$, and $R = \{R_h\}_{h=1}^H$ is the collection of reward functions where each $R_h : \mathcal{S} \times \mathcal{A} \mapsto [0, 1]$. We use $\Delta(\mathcal{S})$ to note the probability simplex on $\mathcal{S}$ (i.e. the space of probability measures with support on $\mathcal{S}$).

We consider a model-based perspective of reinforcement learning, and we denote $\mathcal{P} = \{P(\cdot|\cdot, \cdot) : \mathcal{S} \times \mathcal{A} \mapsto \Delta(\mathcal{S})\}$ as the space of all transition kernels. The space $\mathcal{P}_{\mathrm{M}} \subseteq \mathcal{P}$ of the RMDP is a realizable model space which contains the transition kernel $P^\star$, i.e., $P_h^\star \in \mathcal{P}_{\mathrm{M}}$ for any step $h \in [H]$. Finally, the RMDP is equipped with a mapping $\mathbf{\Phi} : \mathcal{P}_{\mathrm{M}} \mapsto 2^{\mathcal{P}}$ that characterizes the *robust set* of any transition kernel in $\mathcal{P}_{\mathrm{M}}$. Formally, for any transition kernel $P \in \mathcal{P}_{\mathrm{M}}$, we call $\mathbf{\Phi}(P)$ the *robust set* of $P$. One can interpret the transition kernel $P_h^\star \in \mathcal{P}_{\mathrm{M}}$ as the transition kernel of the training environment, while $\mathbf{\Phi}(P_h^\star)$ contains all the possible transition kernels of the test environment.

**Remark 2.1.** *The mapping $\mathbf{\Phi}$ is defined on the realizable model space $\mathcal{P}_{\mathrm{M}}$, while for generality we allow the image of $\mathbf{\Phi}$ to be outside of $\mathcal{P}_{\mathrm{M}}$. That is, a $\widetilde{P} \in \mathbf{\Phi}(P)$ for some $P \in \mathcal{P}_{\mathrm{M}}$ might be in $\mathcal{P}_{\mathrm{M}}^c$.*

**Policy and robust value function.** Given an RMDP $(\mathcal{S}, \mathcal{A}, H, P^\star, R, \mathcal{P}_{\mathrm{M}}, \mathbf{\Phi})$, we consider using a Markovian policy to make decision. A Markovian policy $\pi$ is defined as $\pi = \{\pi_h\}_{h=1}^H$ with $\pi_h : \mathcal{A} \mapsto \Delta(\mathcal{S})$ for each step $h \in [H]$. For simplicity, we use *policy* to refer to a Markovian policy.

Given any policy $\pi$, we define the *robust value function* of $\pi$ with respect to any set of transition kernels $P = \{P_h\}_{h=1}^H \subseteq \mathcal{P}_{\mathrm{M}}$ as the following, for each step $h \in [H]$,

$$V_{h,P,\mathbf{\Phi}}^\pi(s) = \inf_{\widetilde{P}_h \in \mathbf{\Phi}(P_h), 1 \le h \le H} V_h^\pi(s; \{\widetilde{P}_h\}_{h=1}^H), \quad \forall s \in \mathcal{S}, \tag{2.1}$$

$$Q_{h,P,\mathbf{\Phi}}^\pi(s,a) = \inf_{\widetilde{P}_h \in \mathbf{\Phi}(P_h), 1 \le h \le H} Q_h^\pi(s,a; \{\widetilde{P}_h\}_{h=1}^H), \quad \forall(s,a) \in \mathcal{S} \times \mathcal{A}. \tag{2.2}$$

Here $V_h^\pi(\cdot; \{\widetilde{P}_h\}_{h=1}^H)$ and $Q_h^\pi(\cdot; \{\widetilde{P}_h\}_{h=1}^H)$ are the *state-value function* and the *action-value function* [52] of policy $\pi$ in the standard episodic MDP $(\mathcal{S}, \mathcal{A}, H, \{\widetilde{P}_h\}_{h=1}^H, R)$,

$$V_h^\pi(s; \{\widetilde{P}_h\}_{h=1}^H) = \mathbb{E}_{\{\widetilde{P}_h\}_{h=1}^H, \pi}\left[\sum_{i=h}^H R_i(s_i, a_i) \middle| s_h = s\right], \quad \forall s \in \mathcal{S}, \tag{2.3}$$

$$Q_h^\pi(s,a; \{\widetilde{P}_h\}_{h=1}^H) = \mathbb{E}_{\{\widetilde{P}_h\}_{h=1}^H, \pi}\left[\sum_{i=h}^H R_i(s_i, a_i) \middle| s_h = s, a_h = a\right], \quad \forall(s,a) \in \mathcal{S} \times \mathcal{A}, \tag{2.4}$$

where the expectation $\mathbb{E}_{\{\widetilde{P}_h\}_{h=1}^H, \pi}[\cdot]$ is taken with respect to the trajectories induced by the transition kernel $\{\widetilde{P}_h\}_{h=1}^H$ and the policy $\pi$. Intuitively, the robust value function of a policy $\pi$ given transition kernel $P$ is defined as the least expected cumulative reward achieved by $\pi$ when the transition kernel varies in the robust set of $P$. This is how an RMDP takes the perturbed models into consideration.

**$\mathcal{S} \times \mathcal{A}$-rectangular robust set and robust Bellman equation.** Ideally, we would like to consider robust value function that has recursive expressions, just like the Bellman equation satisfied by (2.3) in a standard MDP [52]. To this end, we focus on a generally adopted kind of robust set in our unified framework, which is called the $\mathcal{S} \times \mathcal{A}$-*rectangular robust set* [13].

**Assumption 2.2** ($\mathcal{S} \times \mathcal{A}$-rectangular robust set)**.** *We assume that the mapping $\mathbf{\Phi}$ induces $\mathcal{S} \times \mathcal{A}$-rectangular robust sets. Specifically, the mapping $\mathbf{\Phi}$ satisfies, for $\forall P \in \mathcal{P}_{\mathrm{M}}$,*

$$\mathbf{\Phi}(P) = \bigotimes_{(s,a) \in \mathcal{S} \times \mathcal{A}} \mathcal{P}_\rho(s,a; P), \quad \mathcal{P}_\rho(s,a; P) = \left\{\widetilde{P}(\cdot) \in \Delta(\mathcal{S}) : D(\widetilde{P}(\cdot) \| P(\cdot|s,a)) \le \rho\right\},$$

*for some (pseudo-) distance $D(\cdot \| \cdot)$ on $\Delta(\mathcal{S})$ and some $\rho \in \mathbb{R}_+$. Intuitively, $\mathcal{S} \times \mathcal{A}$-rectangular requires that $\mathbf{\Phi}(P)$ gives decoupled robust sets for $P(\cdot|s,a)$ across different $(s,a)$-pairs. The (pseudo-)distance $D(\cdot \| \cdot)$ can be chosen as a $\phi$-divergence [68, 10] or a $p$-Wasserstein-distance [35].*

Thanks to the $\mathcal{S} \times \mathcal{A}$-rectangular assumption on the mapping $\Phi$, the robust value functions (2.1) of any policy $\pi$ then satisfy a recursive expression, which is called robust Bellman equation [13, 36].

**Proposition 2.3** (Robust Bellman equation). *Under Assumption 2.2, for any $P = \{P_h\}_{h=1}^{H}$ where $P_h \in \mathcal{P}_{\mathrm{M}}$ and any $\pi = \{\pi_h\}_{h=1}^{H}$ with $\pi_h : \mathcal{S} \mapsto \Delta(\mathcal{A})$, the following robust Bellman equation holds,*

$$V_{h,P,\Phi}^{\pi}(s) = \mathbb{E}_{a \sim \pi_h(\cdot|s)}[Q_{h,P,\Phi}^{\pi}(s,a)], \quad \forall s \in \mathcal{S}, \tag{2.5}$$

$$Q_{h,P,\Phi}^{\pi}(s,a) = R_h(s,a) + \inf_{\widetilde{P}_h \in \Phi(P_h)} \mathbb{E}_{s' \sim \widetilde{P}_h(\cdot|s,a)}[V_{h+1,P,\Phi}^{\pi}(s')], \quad \forall(s,a) \in \mathcal{S} \times \mathcal{A}. \tag{2.6}$$

To be self-contained, in Appendix B we provide a detailed proof of the robust Bellman equation in our framework under Assumption 2.2. Equation (2.5) actually says that the infimum over all the transition kernels (recall the definition of $V_{h,P,\Phi}^{\pi}$ in (2.1)) can be decomposed into a "one-step" infimum over the current transition kernel, i.e., $\inf_{\widetilde{P}_h \in \Phi(P_h)}$, and an infimum over the future transition kernels, i.e., $V_{h+1,P,\Phi}^{\pi}$. Such a property is crucial to the algorithmic design and theoretical analysis for RMDPs.

## 2.2 Examples of Robust Markov Decision Processes

In this section, we give concrete examples for the general RMDP framework proposed in Section 2.1. Most existing works on RMDPs hinge on the finiteness assumption on the state space, which fails to deal with prohibitively large or even infinite state space. In our framework, RMDPs can be considered in the paradigm of infinite state space, for which we adopt various powerful function approximation tools including kernel and neural functions. Also, we introduce a new type of RMDP named robust factored MDP, which is a robust extension of standard factored MDPs [17].

**Remark 2.4.** *Besides $\mathcal{S} \times \mathcal{A}$-rectangular-type robust sets (Assumption 2.2), our unified framework of RMDP can also cover other types of robust sets considered in some previous works as special cases, including $\mathcal{S}$-rectangular robust set [61] and $d$-rectangular robust set for linear MDPs [29]. See Section A for a discussion about these two types of robust sets.*

In the sequel, we introduce concrete examples of our framework of RMDP.

**Example 2.5** ($\mathcal{S} \times \mathcal{A}$-rectangular robust tabular MDP). *When the state space $\mathcal{S}$ is a finite set, we call the corresponding model an $\mathcal{S} \times \mathcal{A}$-rectangular robust tabular MDP. Recently, there is a line of works on the $\mathcal{S} \times \mathcal{A}$-rectangular robust tabular MDP [77, 68, 39, 27, 47, 40, 8, 10, 35, 58, 69, 66, 7]. For $\mathcal{S} \times \mathcal{A}$-rectangular robust tabular MDPs, we choose $\mathcal{P}_{\mathrm{M}} = \mathcal{P}$ containing all possible transitions.*

**Remark 2.6.** *We point out that our framework of RMDP under $\mathcal{S} \times \mathcal{A}$-rectangular assumption covers substantially more model than $\mathcal{S} \times \mathcal{A}$-rectangular robust tabular MDP since our state space $\mathcal{S}$ can be infinite. The model space $\mathcal{P}_{\mathrm{M}}$ can be adapted to function approximation methods to handle the infinite state space. Thus any efficient algorithm developed for our framework of RMDPs **can not** be covered by algorithms for $\mathcal{S} \times \mathcal{A}$-rectangular robust tabular MDPs. Example 2.7 and 2.8 are infinite state space $\mathcal{S} \times \mathcal{A}$-rectangular robust MDPs with function approximations.*

**Example 2.7** ($\mathcal{S} \times \mathcal{A}$-rectangular robust MDP with kernel function approximations). *We consider an infinite state space $\mathcal{S} \times \mathcal{A}$-rectangular robust MDP whose realizable model space $\mathcal{P}_{\mathrm{M}}$ is in a reproduced kernel Hilbert space (RKHS). Let $\mathcal{H}$ be a RKHS associated with a positive definite kernel $\mathcal{K} : (\mathcal{S} \times \mathcal{A} \times \mathcal{S}) \times (\mathcal{S} \times \mathcal{A} \times \mathcal{S}) \mapsto \mathbb{R}_{+}$ (See Appendix D.3.1 for a review of the basics of RKHS). We denote the feature mapping of $\mathcal{H}$ by $\psi : \mathcal{S} \times \mathcal{A} \times \mathcal{S} \mapsto \mathcal{H}$. With $\mathcal{H}$, an $\mathcal{S} \times \mathcal{A}$-rectangular robust MDP with kernel function approximation is defined as an $\mathcal{S} \times \mathcal{A}$-rectangular robust MDP with*

$$\mathcal{P}_{\mathrm{M}} = \big\{ P(s'|s,a) = \langle \psi(s,a,s'), \boldsymbol{f} \rangle_{\mathcal{H}} : \boldsymbol{f} \in \mathcal{H}, \|f\|_{\mathcal{H}} \le B_{\mathrm{K}} \big\}, \tag{2.7}$$

*for some $B_{\mathrm{K}} > 0$. Here we implicitly identify $P(\cdot|\cdot,\cdot)$ as the density of the corresponding distribution with respect to a proper base measure on $\mathcal{S} \times \mathcal{A} \times \mathcal{S}$.*

**Example 2.8** ($\mathcal{S} \times \mathcal{A}$-rectangular robust MDP with neural function approximations). *We consider an infinite state space $\mathcal{S} \times \mathcal{A}$-rectangular robust MDP whose realizable model space $\mathcal{P}_{\mathrm{M}}$ is parameterized by an overparameterized neural network. We first define a two-layer fully-connected neural network on some $\mathcal{X} \subseteq \mathbb{R}^{d_{\mathcal{X}}}$ as*

$$\mathrm{NN}(\mathbf{x}; \mathbf{W}, \mathbf{a}) = \frac{1}{\sqrt{m}} \sum_{j=1}^{m} a_j \sigma(\mathbf{x}^{\top} \mathbf{w}_j), \quad \forall \mathbf{x} \in \mathcal{X}, \tag{2.8}$$

*where $m \in \mathbb{N}_+$ is the number of hidden units of* NN, $(\mathbf{W}, \mathbf{a})$ *is the parameters given by* $\mathbf{W} = (\mathbf{w}_1, \cdots, \mathbf{w}_m) \in \mathbb{R}^{d \times m}$, $\mathbf{a} = (a_1, \cdots, a_m)^\top \in \mathbb{R}^m$, *and* $\sigma(\cdot)$ *is the activation function. Now we assume that the state space* $\mathcal{S} \subseteq \mathbb{R}^{d_\mathcal{S}}$ *for some* $d_\mathcal{S} \in \mathbb{N}_+$. *Also, we identify actions via one-hot vectors in* $\mathbb{R}^{|\mathcal{A}|}$, *i.e., we represent* $a \in \mathcal{A}$ *by* $(0, \cdots, 0, 1, 0, \cdots, 0)$ *with 1 in the $a$-th coordinate. Let* $\mathcal{X} = \mathcal{S} \times \mathcal{A} \times \mathcal{S}$ *with* $d_\mathcal{X} = 2d_\mathcal{S} + |\mathcal{A}|$. *Then an* $\mathcal{S} \times \mathcal{A}$-*rectangular robust MDP with neural function approximation is defined as an* $\mathcal{S} \times \mathcal{A}$-*rectangular robust MDP with* $\mathcal{P}_\mathrm{M}$ *given by*

$$\mathcal{P}_\mathrm{M} = \left\{ P(s'|s,a) = \mathrm{NN}((s,a,s'); \mathbf{W}, \mathbf{a}^0) : \|\mathbf{W} - \mathbf{W}^0\|_2 \le B_\mathrm{N} \right\}, \tag{2.9}$$

*for some* $B_\mathrm{N} > 0$ *and some fixed* $(\mathbf{W}^0, \mathbf{a}^0)$ *which can be interpreted as the initialization. We refer to Appendix D.4.1 for more details about neural function approximations and analysis techniques.*

**Example 2.9** ($\mathcal{S} \times \mathcal{A}$-rectangular robust factored MDP). *We consider a factored MDP equipped with* $\mathcal{S} \times \mathcal{A}$-*rectangular factored robust set. A standard factored MDP [17] is defined as follows. Let $d$ be an integer and $\mathcal{O}$ be a finite set. The state space $\mathcal{S}$ is factored as $\mathcal{S} = \mathcal{O}^d$. For each $i \in [d]$, $s[i]$ is the $i$-coordinate of $s$ and it is only influenced by $s[\mathrm{pa}_i]$, where $\mathrm{pa}_i \subseteq [d]$. That is, the transition of a factored MDP can be factorized as*

$$P_h^\star(s'|s,a) = \prod_{i=1}^d P_{h,i}^\star(s'[i]|s[\mathrm{pa}_i], a).$$

*Here we let the realizable model space $\mathcal{P}_\mathrm{M}$ consist of all the factored transition kernels, i.e.,*

$$\mathcal{P}_\mathrm{M} = \left\{ P(s'|s,a) = \prod_{i=1}^d P_i(s'[i]|s[\mathrm{pa}_i], a) \, : \, P_i : \mathcal{S}[\mathrm{pa}_i] \times \mathcal{A} \mapsto \Delta(\mathcal{O}), \forall i \in [d] \right\}.$$

*For an* $\mathcal{S} \times \mathcal{A}$-*rectangular robust factored MDP, we define $\mathbf{\Phi}$ as, for any transition kernel $P(s'|s,a) = \prod_{i=1}^d P_i(s'[i]|s[\mathrm{pa}_i], a) \in \mathcal{P}_\mathrm{M}$, $\mathbf{\Phi}(P) = \bigotimes_{(s,a) \in \mathcal{S} \times \mathcal{A}} \mathcal{P}_{\mathrm{Fac}, \rho}(s, a; P)$, with*

$$\mathcal{P}_{\mathrm{Fac}, \rho}(s, a; P) = \left\{ \prod_{i=1}^d \widetilde{P}_i(\cdot) : \widetilde{P}_i(\cdot) \in \Delta(\mathcal{O}), D(\widetilde{P}_i(\cdot) \| P_i(\cdot | s[\mathrm{pa}_i], a)) \le \rho_i, \forall i \in [d] \right\}.$$

*for some (pseudo-)distance $D$ on $\Delta(\mathcal{O})$ and some positive real numbers $\{\rho_i\}_{i=1}^d$.*

## 2.3 Offline Reinforcement Learning in Robust Markov Decision Processes

In this section, we define the offline RL protocol in a RMDP $(\mathcal{S}, \mathcal{A}, H, P^\star, R, \mathcal{P}_\mathrm{M}, \mathbf{\Phi})$. The learner is given the realizable model space $\mathcal{P}_\mathrm{M}$ and the robust mapping $\mathbf{\Phi}$, but the learner doesn't know the transition kernel $P^\star$. For simplicity, we assume that the learner knows the reward function $R$[2].

**Offline dataset generation.** We assume that the learner is given an offline dataset $\mathbb{D}$ that consists of $n$ i.i.d. trajectories generated from the standard MDP $(\mathcal{S}, \mathcal{A}, H, P^\star, R)$ using some behavior policy $\pi^\mathrm{b}$. For each $\tau \in [n]$, the trajectory has the form of $\{(s_h^\tau, a_h^\tau, r_h^\tau)\}_{h=1}^H$, satisfying that $a_h^\tau \sim \pi_h^\mathrm{b}(\cdot | s_h^\tau)$, $r_h^\tau = R_h(s_h^\tau, a_h^\tau)$, and $s_{h+1}^\tau \sim P_h^\star(\cdot | s_h^\tau, a_h^\tau)$ for each step $h \in [H]$.

Given transition kernels $P = \{P_h\}_{h=1}^H$ and a policy $\pi$, we use $d_{P,h}^\pi(\cdot, \cdot)$ to denote the state-action visitation distribution at step $h$ when following policy $\pi$ and transition kernel $P$. With this notation, the distribution of $(s_h^\tau, a_h^\tau)$ can be written as $d_{P^\star, h}^{\pi^\mathrm{b}}$ or simply $d_{P^\star, h}^\mathrm{b}$, for each $\tau \in [n]$ and $h \in [H]$. We also use $d_{P^\star, h}^{\pi^\mathrm{b}}(\cdot)$ to denote the marginal distribution of state at step $h$ when there is no confusion.

**Learning objective.** In offline robust RL, the goal is to learn the policy $\pi^\star$ from the offline dataset $\mathbb{D}$ which maximizes the robust value function $V_{1, P^\star, \mathbf{\Phi}}^\pi$, that is,

$$\pi^\star = \operatorname*{argsup}_{\pi \in \Pi} V_{1, P^\star, \mathbf{\Phi}}^\pi(s_1), \quad s_1 \in \mathcal{S}, \tag{2.10}$$

where $\Pi = \{\pi = \{\pi_h\}_{h=1}^H \,|\, \pi_h : \mathcal{S} \mapsto \Delta(\mathcal{A})\}$ denotes the collection of all Markovian policies. In view of (2.10), we call $\pi^\star$ the *optimal robust policy*. Equivalently, we want to learn a policy $\widehat{\pi} \in \Pi$ which minimizes the suboptimality gap between $\widehat{\pi}$ and $\pi^\star$, defined as[3]

$$\mathrm{SubOpt}(\widehat{\pi}; s_1) := V_{1, P^\star, \mathbf{\Phi}}^{\pi^\star}(s_1) - V_{1, P^\star, \mathbf{\Phi}}^{\widehat{\pi}}(s_1), \quad \forall s_1 \in \mathcal{S}. \tag{2.11}$$

---

[2]This is reasonable since learning the reward function is easier than learning the transition kernel.

[3]Without loss of generality, we assume that the initial state is fixed to some $s_1 \in \mathcal{S}$. Our algorithm and theory can be directly extended to the case when $s_1 \sim \rho \in \Delta(\mathcal{S})$.

---

**Algorithm 1** Doubly Pessimistic Model-based Policy Optimization (P$^2$MPO)

---

1: **Input**: model space $\mathcal{P}_{\mathrm{M}}$, mapping $\mathbf{\Phi}$, dataset $\mathbb{D}$, policy class $\Pi$, algorithm `ModelEst`.
2: Model estimation step:
3: Obtain a confidence region $\widehat{\mathcal{P}} = \texttt{ModelEst}(\mathbb{D}, \mathcal{P}_{\mathrm{M}})$.
4: Doubly pessimistic policy optimization step:
5: Set policy $\widehat{\pi}$ as $\mathrm{argsup}_{\pi \in \Pi} J_{\texttt{Pess}^2}(\pi)$, where $J_{\texttt{Pess}^2}(\pi)$ is defined in (3.1).
6: **Output**: $\widehat{\pi} = \{\widehat{\pi}_h\}_{h=1}^H$.

---

## 3 Algorithm: Generic Framework and Unified Theory

In this section, we propose Doubly Pessimistic Model-based Policy Optimization (P$^2$MPO) algorithm to solve offline RL in the RMDP framework we introduce in Section 2.1, and we establish a unified theoretical guarantee for P$^2$MPO. Our proposed algorithm and theory show that *double pessimism* is a general principle for designing efficient algorithms for offline robust RL. The algorithm features three key points: i) learning the optimal robust policy $\pi^\star$ approximately; ii) requiring only a partial coverage property of the offline dataset $\mathbb{D}$; iii) able to handle infinite state space via function approximations.

We first introduce our proposed algorithm framework P$^2$MPO in Section 3.1. Then we establish a unified analysis for P$^2$MPO in Section 3.2. Our algorithm framework can be specified to solve all the concrete examples of RMDP we introduce in Section 2.2, which we show in Section 4.

### 3.1 Algorithm Framework: P$^2$MPO

The P$^2$MPO algorithm framework (Algorithm 1) consists of a *model estimation step* and a *doubly pessimistic policy optimization step*, which we introduce in the following respectively.

**Model estimation step (Line 3).** The P$^2$MPO algorithm framework first constructs an estimation of the transition kernels $P^\star = \{P_h^\star\}_{h=1}^H$, i.e., it estimates the dynamic of the training environment. It implements a sub-algorithm `ModelEst`$(\mathbb{D}, \mathcal{P}_{\mathrm{M}})$ that returns a confidence region $\widehat{\mathcal{P}}$ for $P^\star = \{P_h^\star\}_{h=1}^H$. Specifically, $\widehat{\mathcal{P}} = \{\widehat{\mathcal{P}}_h\}_{h=1}^H$ with $\widehat{\mathcal{P}}_h \subseteq \mathcal{P}_{\mathrm{M}}$ for each step $h \in [H]$.

The sub-algorithm `ModelEst` can be tailored to specific RMDPs. We refer to Section 4 for detailed implementations of `ModelEst` for different examples of RMDPs introduced in Section 2.2. Ideally, to ensure sample-efficient learning, we need $\widehat{\mathcal{P}} = \texttt{ModelEst}(\mathbb{D}, \mathcal{P}_{\mathrm{M}})$ to satisfy: i) the transition kernels $P^\star = \{P_h^\star\}_{h=1}^H$ are contained in $\widehat{\mathcal{P}} = \{\widehat{\mathcal{P}}_h\}_{h=1}^H$; ii) each transition kernel $P_h \in \widehat{\mathcal{P}}_h$ enjoys a small "robust estimation error" which is highly related to the robust Bellman equation in (2.5). We quantify these two conditions of $\widehat{\mathcal{P}}$ for sample-efficient learning in Section 3.2.

**Doubly pessimistic policy optimization step (Line 5).** After **model estimation step**, P$^2$MPO performs policy optimization to find the optimal robust policy. To learn the optimal robust policy in the face of uncertainty, P$^2$MPO adopts a *double pessimism* principle. To explain, this general principle has two sources of pessimism: i) pessimism in the face of data uncertainty; ii) pessimism to find a robust policy. Specifically, for any policy, we first estimate its robust value function via two infimums, where one is an infimum over the confidence set constructed in the model estimation step, and one is an infimum over the robust sets. Formally, for any $\pi \in \Pi$, we define the doubly pessimistic estimator

$$J_{\texttt{Pess}^2}(\pi) = \inf_{P_h \in \widehat{\mathcal{P}}_h, 1 \leq h \leq H} \inf_{\widetilde{P}_h \in \mathbf{\Phi}(P_h), 1 \leq h \leq H} V_1^\pi(s_1; \{\widetilde{P}_h\}_{h=1}^H), \qquad (3.1)$$

where $V_1^\pi$ is the standard value function of policy $\pi$ defined in (2.3). Then P$^2$MPO outputs the policy $\widehat{\pi}$ that maximizes the doubly pessimistic estimator $J_{\texttt{Pess}^2}(\pi)$ defined in (3.1), i.e.,

$$\widehat{\pi} = \mathrm{argsup}_{\pi \in \Pi} J_{\texttt{Pess}^2}(\pi). \qquad (3.2)$$

The novelty of the doubly pessimistic policy optimization step is performing pessimism from the two sources (data uncertainty and robust optimization) simultaneously. Compared with the previous works on standard offline RL in MDPs [62, 54] and offline RL in RMDPs without pessimism [68, 77, 40], they only contain one source of pessimism in algorithm design, contrasting with our algorithm.

We note that a recent work [47] also studied robust offline RL in $\mathcal{S} \times \mathcal{A}$-rectangular robust tabular MDPs (Example 2.5) using pessimism techniques. Compared with our double pessimism principle,

their algorithm performs pessimism in face of data uncertainty i) depending on the tabular structure of the model since a point-wise pessimism penalty is needed and ii) depending on the specific form of the robust set $\mathbf{\Phi}(P)$, which makes it difficult to adapt to the infinite state space case with general function approximations and general types of robust set $\mathbf{\Phi}(P)$.

## 3.2 Unified Theoretical Analysis

In this section, we establish a unified theoretical analysis for the $\texttt{P}^2\texttt{MPO}$ algorithm framework proposed in Section 3.1. We first specify the two conditions that the **model estimation step** of $\texttt{P}^2\texttt{MPO}$ should satisfy in order for sample-efficient learning. Then we establish an upper bound of suboptimality of the policy obtained by $\texttt{P}^2\texttt{MPO}$ given that these two conditions are satisfied. In Section 4, we show that the specific implementations of the sub-algorithm $\texttt{ModelEst}$ for the RMDPs examples in Section 2.2 satisfy these two conditions, which results in tailored suboptimality bounds for these examples.

**Conditions.** The two conditions on the **model estimation step** are given by the following.

**Condition 3.1** ($\delta$-accuracy). With probability at least $1 - \delta$, it holds that $P_h^\star \in \widehat{\mathcal{P}}_h$ for any $h \in [H]$.

**Condition 3.2** ($\delta$-model estimation error). With probability at least $1 - \delta$, it holds that

$$\mathbb{E}_{(s,a)\sim d_{P^\star,h}^{\mathrm{b}}} \left[ \left( \inf_{\widetilde{P}_h \in \mathbf{\Phi}(P_h)} \widetilde{\mathbb{P}}_h(V_{h+1,P,\mathbf{\Phi}}^{\pi^\star})(s,a) - \inf_{\widetilde{P}_h \in \mathbf{\Phi}(P_h^\star)} \widetilde{\mathbb{P}}_h(V_{h+1,P,\mathbf{\Phi}}^{\pi^\star})(s,a) \right)^2 \right] \leq \mathrm{Err}_h^{\mathbf{\Phi}}(n,\delta).$$

for any $P = \{P_h\}_{h=1}^H$ with $P_h \in \widehat{\mathcal{P}}_h$. Here $\widetilde{\mathbb{P}}_h(V_{h+1,P,\mathbf{\Phi}}^{\pi^\star})(s,a) = \mathbb{E}_{s'\sim\widetilde{P}_h(\cdot|s,a)}[V_{h+1,P,\mathbf{\Phi}}^{\pi^\star}(s')]$

Condition 3.1 requires that the confidence region $\widehat{\mathcal{P}}_h$ contains the transition kernel of the training environment $P_h^\star$ with high probability. Condition 3.2 requires that each transition kernel $P_h \in \widehat{\mathcal{P}}_h$ induces an error from $P_h^\star$ less than certain quantity $\mathrm{Err}_h^{\mathbf{\Phi}}(n,\delta)$, where the error is adapted from the robust Bellman equation (2.5) and involves an infimum over the robust set of $P_h$ and $P_h^\star$. In specific implementations of $\texttt{ModelEst}$ for RMDP examples in Section 4, we show that the quantity $\mathrm{Err}_h^{\mathbf{\Phi}}(n,\delta)$ generally scales with $\widetilde{\mathcal{O}}(n^{-1})$, where $n$ is the number of trajectories in the offline dataset.

**Suboptimality analysis.** Now we establish a unified suboptimality bound for the $\texttt{P}^2\texttt{MPO}$ algorithm framework. Thanks to the double pessimism principle of $\texttt{P}^2\texttt{MPO}$, we can prove a suboptimality bound while only making a mild *robust partial coverage assumption* on the dataset.

**Assumption 3.3** (Robust partial coverage). *We assume that*

$$C_{P^\star,\mathbf{\Phi}}^\star := \sup_{1\leq h\leq H} \sup_{P=\{P_h\}_{h=1}^H, P_h\in\mathbf{\Phi}(P_h^\star)} \mathbb{E}_{(s,a)\sim d_{P^\star,h}^{\mathrm{b}}} \left[ \left( \frac{d_{P,h}^{\pi^\star}(s,a)}{d_{P^\star,h}^{\mathrm{b}}(s,a)} \right)^2 \right] < +\infty,$$

*and we call $C_{P^\star,\mathbf{\Phi}}^\star$ the robust partial coverage coefficient.*

To interpret, Assumption 3.3 only requires that the dataset covers the visitation distribution of the optimal policy $\pi^\star$, but in a robust fashion since $C_{P^\star,\mathbf{\Phi}}^\star$ considers all possible transition kernels in the robust set $\mathbf{\Phi}(P^\star)$. The robust consideration in $C_{P^\star,\mathbf{\Phi}}^\star$ is because in RMDPs the policies are all evaluated in a robust way. This partial-coverage-style assumption is much weaker than full-coverage-style assumptions [68, 77, 40] which require either a uniformly lower bounded dataset distribution or covering the visitation distribution of any $\pi \in \Pi$. For $\mathcal{S} \times \mathcal{A}$-rectangular robust tabular MDPs (Example 2.5), the robust partial coverage coefficient $C_{P^\star,\mathbf{\Phi}}^\star$ is similar with the partial coverage coefficient proposed by [47] who studied tabular RMDPs under partial coverage. We highlight that beyond $\mathcal{S} \times \mathcal{A}$-rectangular robust tabular MDPs, our robust partial coverage assumption can handle other examples of RMDPs (Section 2.2) under our unified theory.

Our main result is the following theorem. See Appendix C for a detailed proof.

**Theorem 3.4** (Suboptimality of $\texttt{P}^2\texttt{MPO}$). *Under Assumptions 2.2 and 3.3, suppose that Algorithm 1 implements a sub-algorithm that satisfies Conditions 3.1 and 3.2, then with probability at least $1 - 2\delta$,*

$$\mathrm{SubOpt}(\widehat{\pi}; s_1) \leq \sqrt{C_{P^\star,\mathbf{\Phi}}^\star} \cdot \sum_{h=1}^H \sqrt{\mathrm{Err}_h^{\mathbf{\Phi}}(n,\delta)}.$$

When $\mathrm{Err}_h^{\mathbf{\Phi}}(n,\delta)$ achieves a rate of $\widetilde{\mathcal{O}}(n^{-1})$, then $\texttt{P}^2\texttt{MPO}$ enjoys a $\widetilde{\mathcal{O}}(n^{-1/2})$-suboptimality. In the following Section 4, we give specific implementations of the model estimation step of $\texttt{P}^2\texttt{MPO}$ for each example of RMDP in Section 2. The implementations will make Conditions 3.1 and 3.2 satisfied and thus specify the unified result Theorem 3.4.

# 4 Implementations of P²MPO for Examples of RMDPs

In this section, we provide concrete implementations of the `ModelEst` sub-algorithm in P²MPO (Algorithm 1). In Section 4.1, we implement `ModelEst` for all the RMDPs that satisfy Assumption 2.2, and we specify the suboptimality bounds in Theorem 3.4 to Examples 2.5, 2.7, 2.8 in Section 2.2. In Section 4.2, we implement `ModelEst` for $\mathcal{S} \times \mathcal{A}$-rectangular robust factored MDPs (Example 2.9) and specify Theorem 3.4 to this example.

## 4.1 Model Estimation for General RMDPs with $\mathcal{S} \times \mathcal{A}$-rectangular Robust Sets

Using the offline data $\mathbb{D}$, we first construct the *maximum likelihood estimator* (MLE) of the transition kernel $P^\star$. Specifically, for each step $h \in [H]$, we define

$$\widehat{P}_h = \arg\max_{P \in \mathcal{P}_{\mathrm{M}}} \frac{1}{n} \sum_{\tau=1}^{n} \log P(s_{h+1}^\tau | s_h^\tau, a_h^\tau). \tag{4.1}$$

After, we construct a confidence region for the MLE estimator, denoted by $\widehat{\mathcal{P}}$. Specifically, $\widehat{\mathcal{P}}$ contains all transitions which have a small total variance distance from $\widehat{P}$. For each step $h \in [H]$, we define

$$\widehat{\mathcal{P}}_h = \left\{ P \in \mathcal{P}_{\mathrm{M}} : \frac{1}{n} \sum_{\tau=1}^{n} \|\widehat{P}_h(\cdot|s_h^\tau, a_h^\tau) - P(\cdot|s_h^\tau, a_h^\tau)\|_1^2 \leq \xi \right\}. \tag{4.2}$$

Here $\xi > 0$ is a tuning parameter that controls the size of the confidence region $\widehat{\mathcal{P}}_h$. Finally, we set `ModelEst`$(\mathbb{D}, \mathcal{P}_{\mathrm{M}}) = \widehat{\mathcal{P}} = \{\widehat{\mathcal{P}}_h\}_{h=1}^{H}$ with $\widehat{\mathcal{P}}_h$ given in (4.2). In the sequel, we mainly consider the distance $D(\cdot\|\cdot)$ in Assumption 2.2 to be KL-divergence and TV-distance. The following corollary specifies Theorem 3.4 to model estimation step given by (4.2). See Appendix D for a detailed proof.

**Corollary 4.1** (Suboptimality of P²MPO: $\mathcal{S} \times \mathcal{A}$-rectangular robust MDP). *Under Assumption 2.2, 3.3, setting the tuning parameter $\xi$ as*

$$\xi = \frac{C_1 \log(C_2 H \mathcal{N}_{[]}(1/n^2, \mathcal{P}_{\mathrm{M}}, \|\cdot\|_{1,\infty})/\delta)}{n},$$

*for some constants $C_1, C_2 > 0$, P²MPO with model estimation step given by (4.2) satisfies that*

♠ *when $D(\cdot\|\cdot)$ is KL-divergence and Assumption D.3 holds with parameter $\underline{\lambda}$, then with probability at least $1 - 2\delta$,*

$$\mathrm{SubOpt}(\widehat{\pi}; s_1) \leq \frac{\sqrt{C_{P^\star, \Phi}^\star} H^2 \exp(H/\underline{\lambda})}{\rho} \cdot \sqrt{\frac{C_1' \log(C_2' H \mathcal{N}_{[]}(1/n^2, \mathcal{P}_{\mathrm{M}}, \|\cdot\|_{1,\infty})/\delta)}{n}}.$$

♠ *when $D(\cdot\|\cdot)$ is TV-divergence, then with probability at least $1 - 2\delta$,*

$$\mathrm{SubOpt}(\widehat{\pi}; s_1) \leq \sqrt{C_{P^\star, \Phi}^\star} H^2 \cdot \sqrt{\frac{C_1' \log(C_2' H \mathcal{N}_{[]}(1/n^2, \mathcal{P}_{\mathrm{M}}, \|\cdot\|_{1,\infty})/\delta)}{n}}.$$

**$\mathcal{S} \times \mathcal{A}$-rectangular robust tabular MDP (Example 2.5).** When $\mathcal{S}$ is finite as in Example 2.5, the MLE estimator (4.1) coincides the empirical estimator

$$\widehat{P}_h(s'|s, a) = \frac{\sum_{\tau=1}^{n} \mathbf{1}\{s_h^\tau = s, a_h^\tau = a, s_{h+1}^\tau = s'\}}{1 \vee \sum_{\tau=1}^{n} \mathbf{1}\{s_h^\tau = s, a_h^\tau = a\}}, \tag{4.3}$$

which is adopted by [77, 68, 39, 47, 40]. Furthermore, in Example 2.5, the realizable model space $\mathcal{P}_{\mathrm{M}} = \{P : \mathcal{S} \times \mathcal{A} \mapsto \Delta(\mathcal{S})\}$. When $\mathcal{S}$ and $\mathcal{A}$ are finite, we can bound the bracket number of $\mathcal{P}_{\mathrm{M}}$ as

$$\log \mathcal{N}_{[]}(1/n^2, \mathcal{P}_{\mathrm{M}}, \|\cdot\|_{1,\infty}) \leq 2|\mathcal{S}|^2 |\mathcal{A}| \log(n). \tag{4.4}$$

Combining (4.4) and Corollary 4.1, we can conclude that: i) under TV-distance the suboptimality of P²MPO for $\mathcal{S} \times \mathcal{A}$-rectangular robust tabular RMDP is given by $\mathcal{O}(H^2 \sqrt{C_{P^\star, \Phi}^\star |\mathcal{S}|^2 |\mathcal{A}| \log(nH/\delta)/n})$, ii) under KL-divergence the suboptimality of P²MPO for $\mathcal{S} \times \mathcal{A}$-rectangular robust tabular MDP is given by $\mathcal{O}(H^2 \exp(H/\underline{\lambda})/\rho \cdot \sqrt{C_{P^\star, \Phi}^\star |\mathcal{S}|^2 |\mathcal{A}| \log(nH/\delta)/n})$. We prove (4.4) in Appendix D.2.

**Remark 4.2.** *We note that for KL-divergence robust sets, the dependence on $\exp(H)$ is due to the usage of general function approximations, which also appears in a recent work [29] for RMDPs with linear function approximations. For the special case of robust tabular MDPs, under KL-divergence, existing work [47] derived sample complexities without $\exp(H)$, but with an additional dependence on $1/d_{\min}^{\mathrm{b}}$ and $1/P_{\min}^{\star}$ Here $d_{\min}^{\mathrm{b}} = \min_{(s,a,h):d_{P^\star,h}^{\pi^{\mathrm{b}}}(s,a)>0} d_{P^\star,h}^{\pi^{\mathrm{b}}}(s,a)$ and $P_{\min}^{\star} = \min_{(s,s',h):P_h(s'|s,\pi_h^\star(s))>0} P_h^\star(s'|s,\pi_h^\star(s))$. We remark that our analysis for $\mathtt{P}^2\mathtt{MPO}$ algorithm can be tailored to the tabular case and become $\exp(H)$-free using their techniques, with the cost of an additional dependence on $1/d_{\min}^{\mathrm{b}}$ and $1/P_{\min}^{\star}$. But we note that in the infinite state space case, both the $1/d_{\min}^{\mathrm{b}}$-dependence and the $1/P_{\min}$-dependence becomes problematic. So, it serves as an interesting future work to answer whether one can derive both $\exp(H)$-free and $(1/d_{\min}^{\mathrm{b}}, 1/P_{\min}^{\star})$-free results for (general) function approximations under KL-divergence.*

$\mathcal{S} \times \mathcal{A}$**-rectangular robust MDP with kernel and neural function approximations (Examples 2.7 and 2.8).** By specifying the bracket numbers in Corollary 4.1, we can provide the detailed suboptimality guarantees for $\mathcal{S} \times \mathcal{A}$-rectangular robust MDP with kernel and neural function approximations. Due to space limitations, we defer the detailed results to Appendices D.3 and D.4.

## 4.2 Model Estimation for $\mathcal{S} \times \mathcal{A}$-rectangular Robust Factored MDPs (Example 2.9)

We first construct MLE estimator for each factor $P_{h,i}^\star$ of the transition $P_h^\star = \prod_{i=1}^d P_{h,i}^\star$, that is,

$$\widehat{P}_{h,i} = \underset{P_i:\mathcal{S}[\mathrm{pa}_i]\times\mathcal{A}\mapsto\Delta(\mathcal{O})}{\arg\max} \frac{1}{n} \sum_{k=1}^n \log P(s_{h+1}^\tau[i]|s_h^\tau[\mathrm{pa}_i], a_h^\tau). \tag{4.5}$$

Then given $\{\widehat{P}_{h,i}\}_{i=1}^d$ we construct a confidence region that is factored across $i \in [d]$. Specifically,

$$\widehat{\mathcal{P}}_h = \left\{ P(s'|s,a) = \prod_{i=1}^d P_i(s'[i]|s[\mathrm{pa}_i],a) : \frac{1}{n} \sum_{i=1}^n \|(P_i - \widehat{P}_{h,i})(\cdot|s_h^\tau[\mathrm{pa}_i], a_h^\tau)\|_1^2 \leq \xi_i, \forall i \right\}. \tag{4.6}$$

Finally, we set $\mathtt{ModelEst}(\mathbb{D}, \mathcal{P}_{\mathrm{M}}) = \widehat{\mathcal{P}} = \{\widehat{\mathcal{P}}\}_{h=1}^H$ with $\widehat{\mathcal{P}}_h$ given in (4.6). The following corollary specifies Theorem 3.4 to model estimation step given by (4.6). See Appendix E for a detailed proof.

**Corollary 4.3** (Suboptimality of $\mathtt{P}^2\mathtt{MPO}$: $\mathcal{S} \times \mathcal{A}$-rectangular robust factored MDP). *Supposing the RMDP is an $\mathcal{S} \times \mathcal{A}$-rectangular robust factored MDP, under the same Assumptions and parameter choice in Theorem 3.4 and Proposition E.1, $\mathtt{P}^2\mathtt{MPO}$ with model estimation step given by (4.6) satisfies*

♣ *when $D(\cdot\|\cdot)$ is KL-divergence and Assumption D.3 holds with parameter $\underline{\lambda}$, then with probability at least $1 - 2\delta$, (defining $\rho_{\min} = \min_{i\in[d]} \rho_i$)*

$$\mathrm{SubOpt}(\widehat{\pi}; s_1) \leq \frac{\sqrt{C_{P^\star,\boldsymbol{\Phi}}^\star} H^2 \exp(H/\underline{\lambda})}{\rho_{\min}} \cdot \sqrt{\frac{dC_1' \sum_{i=1}^d |\mathcal{O}|^{1+|\mathrm{pa}_i|} |\mathcal{A}| \log(C_2'nd/\delta)}{n}}.$$

♣ *when $D(\cdot\|\cdot)$ is TV-divergence, then with probability at least $1 - 2\delta$,*

$$\mathrm{SubOpt}(\widehat{\pi}; s_1) \leq \sqrt{C_{P^\star,\boldsymbol{\Phi}}^\star} H^2 \sqrt{\frac{dC_1' \sum_{i=1}^d |\mathcal{O}|^{1+|\mathrm{pa}_i|} |\mathcal{A}| \log(C_2'nd/\delta)}{n}}.$$

Compared with the suboptimality bounds for $\mathcal{S} \times \mathcal{A}$-rectangular robust MDPs in Section 4.1, the suboptimality of $\mathcal{S} \times \mathcal{A}$-rectangular robust factored MDPs with $\mathtt{ModelEst}$ given in (4.6) only scales with $\sum_{i=1}^d |\mathcal{O}|^{1+|\mathrm{pa}_i|}$ instead of scaling with $|\mathcal{S}| = \prod_{i=1}^d |\mathcal{O}|$ which is of order $\exp(d)$. This justifies the benefit of considering $\mathcal{S} \times \mathcal{A}$-rectangular robust factored MDPs when the transition kernels of training and testing environments enjoy factored structures.

# 5 Conclusion and Discussions

This paper proposes a general learning principle — double pessimism — for robust offline RL. Based on this learning principle, we propose a generic algorithm that only requires robust partial coverage data to solve $\mathcal{S} \times \mathcal{A}$-rectangular RMDPs with general function approximation. Our results are ready to be extended to $d$-rectangular linear RMDPs [29]. See Appendix A for details. In Appendix A, we also provide some challenges to perform sample efficient RL in $\mathcal{S}$-rectangular RMDPs.

## Acknowledgments and Disclosure of Funding

The material in this paper is based upon work supported by the Air Force Office of Scientific Research under award number FA9550-20-1-0397. Additional support is gratefully acknowledged from NSF 1915967, 2118199, 2229012, 2312204.

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
