# OpenReview forum: "Double Pessimism is Provably Efficient for Distributionally Robust Offline Reinforcement Learning: Generic Algorithm and Robust Partial Coverage"
_NeurIPS.cc/2023/Conference — NeurIPS 2023 poster_

### Official Review · Reviewer_Ty5v · 2023-06-10

**Soundness:** 3 good
**Presentation:** 3 good
**Contribution:** 3 good
**Rating:** 6
**Confidence:** 3

**Summary:**

This paper proposed a unified framework of Robust Markov Decision Process, which includes many newly proposed models as special cases. Under their generic framework, they proposed Doubly Pessimistic Model-based Policy Optimization (P^2MPO) that adopts a double pessimism principle for policy optimization. They showed that the suboptimality gap of the proposed algorithm can be upper bounded by the model estimation error and the robust partial coverage coefficient. They further provided concrete implementations of their algorithm on some specific models and showed that their algorithm enjoys a $n^{-1/2}$ convergence rate with $n$ being the number of trajectories in the offline dataset.

**Strengths:**

- They proposed a generic framework that covers many models such as $\mathcal{S}\times\mathcal{A}$-rectangular tabular RMDPs, $\mathcal{S}\times\mathcal{A}$-rectangular kernel RMDPs, $\mathcal{S}\times\mathcal{A}$-rectangular neural RMDPs, $\mathcal{S}\times\mathcal{A}$-rectangular factored RMDPs.
- Instead of covering the visitation distribution of any policy, they only require a partial coverage style assumption, i.e., the dataset covers the visitation distribution of the optimal policy in a robust fashion.


**Weaknesses:**

- Computationally inefficient, i.e., not sure how to solve (4.1), (4.2) and (3.2) efficiently.
- Though the authors showed how the model estimation step is implemented on some specific models, it is not clear how to implement the model estimation step that satisfies Condition 3.1 and Condition 3.2 in general.


**Questions:**

- Why the robust partial coverage assumption (which considers all the possible transition kernels in a robust set) is much weaker than assuming the dataset covers the visitation distribution of any possible policy (under only the true transition kernel)? In other words, does the full-coverage-style assumption implies the partial-coverage-style assumption?
- How does the rates obtained in Section 4 Corollary 4.1, 4.3 compare with the previous works?


**Limitations:**

See weaknesses

---

> ### Author Rebuttal · Authors · 2023-08-08
>
> **Response to Reviewer Ty5v**
>
> Thanks very much for your appreciation of our work!  In the following, we will try our best to address all your concerns and questions.
>
> **Q1: The algorithm is computationally inefficient, i.e., not sure how to solve (4.1), (4.2) and (3.2) efficiently.**
>
> **A1:** We clarify that our work focuses on the statistical side of robust offline RL under general function approximations, which remains an important open problem. Our algorithm is therefore information-theoretic and is indeed intractable with an abstract function approximation class. The objective we designed is to achieve statistical efficiency in the most general setup. Computational efficiency is not our focus.
>
> Meanwhile, if we specify the model space, for example, tabular case or linear/kernel function classes, we can design approximations of our doubly pessimistic algorithm. For example in tabular RMDPs, we can replace the iterative infimum in our double pessimism objective (3.2) by an LCB-style bonus. This has been shown to be efficient both theoretically and experimentally by [1], which can demonstrate the efficiency and practicality of our general algorithm design.
>
> **Q2: Though the authors showed how the model estimation step is implemented on some specific models, it is not clear how to implement the model estimation step that satisfies Condition 3.1 and Condition 3.2 in general.**
>
> **A2:** Conditions 3.1 and 3.2 involved in our unified theory are guidelines for customizing the model estimation subroutine for different RMDP examples, which shows the flexibility of our algorithm and theory. Actually, different RMDP examples have different structures. By tailoring the model estimation step for specific RMDP examples, one can utilize their structures to the best and obtain better sample efficiency.
>
> For example, for $\mathcal{S}\times\mathcal{A}$-rectangular robust factored MDP (Example 2.9) with finite states and actions, our proposed model estimation subroutine (Section 4.2) enjoys better sample efficiency than naively applying the model estimation step designed for general $\mathcal{S}\times\mathcal{A}$-rectangular robust tabular MDPs (Example 2.5 in Section 4.1). This is also the case for $d$-rectangular robust linear MDPs (Appendix B).
>
> In general, an RMDP instance could have a very complicated and detailed structure. Designing an once-for-all model estimation subroutine would lose certain information which may help to boost the sample-efficiency. Therefore, implementing the model estimation step that satisfies Conditions 3.1 and 3.2 in general is not our focus.
>
>
>
> **Q3: Why the robust partial coverage assumption (which considers all the possible transition kernels in a robust set) is much weaker than assuming the dataset covers the visitation distribution of any possible policy (under only the true transition kernel)? In other words, does the full-coverage-style assumption implies the partial-coverage-style assumption?**
>
> **A3:** Thanks for pointing that out! We clarify this point in the following.
>
> In our paper (Line 262 to 264) we mentioned two types of coverage condition: (i) the distribution of the data is uniformly lower bounded; (ii) covering the visitation distribution of any $\pi\in\Pi$. It seems that we did not elaborate under what transition kernels (ii) holds. By (ii), we meant to refer to the robust-full-coverage-style assumption which holds for all $\pi\in\Pi$ and $P\in\boldsymbol{\Phi}(P^{\star})$, parallel to our robust-partial-coverage-style assumption in terms of transition kernels. When considering the full-coverage-style assumption you mentioned, it is important to note that a direct implication between it and our robust-partial-coverage-style assumption cannot be established. However, as is shown by an information-theoretical lower bound by [1] for the tabular RMDP case, the robust-partial-coverage-style assumption is the "minimal" assumption on the offline data to some extent.
>
> Meanwhile, we remark that all previous full-coverage-style offline setup papers use (i) as their data assumption. Since (i) requires that the distribution of the state-action pairs are uniformly lower bounded, this kind of full-coverage assumption is definitely much stronger than our robust partial coverage assumption.
>
>
> **Q4: How does the rates obtained in Section 4 Corollary 4.1 & 4.3 compare with the previous works?**
>
> **A4:** We have compared our results in Corrolary 4.1 with existing work [1] in Remark 4.2. Regarding our results for robust factored MDPs in Corollary 4.3, their is no existing work to compare with. This novel model is proposed by our work, and our algorithm stands as the first efficient algorithm designed for this problem.
>
> **References:**
>
> [1] Shi, Laixi, and Yuejie Chi. "Distributionally robust model-based offline reinforcement learning with near-optimal sample complexity." arXiv preprint arXiv:2208.05767 (2022).

---

> > ### Comment · Reviewer_Ty5v · 2023-08-12
> > **Thanks for the rebuttal**
> >
> > I have read the rebuttal and all the other reviews. I think the paper has novel contributions (i.e., double pessimism, P^2MPO) but my concern is still the computational issue. Thus I will keep my score.

---

> > > ### Author Response · Authors · 2023-08-13
> > >
> > > Thank you much very for your appreciation of the contributions and novelties of our work! We will keep improving the paper following your questions and suggestions in the revision.

---

### Official Review · Reviewer_F7Eb · 2023-06-26

**Soundness:** 3 good
**Presentation:** 2 fair
**Contribution:** 3 good
**Rating:** 7
**Confidence:** 4

**Summary:**

This study focuses on distributionally robust offline reinforcement learning to discover an optimal robust policy using an offline dataset for effective performance in perturbed environments. A novel algorithm framework called Doubly Pessimistic Model-based Policy Optimization (P2MPO) is proposed. P2MPO combines a flexible model estimation subroutine with a doubly pessimistic policy optimization step, leveraging the double pessimism principle to address challenges arising from behavior-policy mismatch and model perturbations. The study demonstrates the sample efficiency of P2MPO with robust partial coverage data and highlights its convergence rate. It introduces the concept of double pessimism as a general learning principle for robust offline RL, showcasing its efficiency in kernel and neural function approximators.

**Strengths:**

I think this work really pushes the robust RL community research efforts further by answering:
> can we design a generic algorithm for robust offline RL in the context of function approximation?

The main contribution of double pessimism: one for robustness of dynamics uncertainty and one for model-estimation using offline data is a really nice idea worthy for publication at NeurIPS.

**Weaknesses:**

I have only minor weaknesses for this work as follows:

1. The robust Bellman equation for d-rectangular sets are not formally proven. The proof isn't there even in [Ma et al. 22] to the extent of my search. The current Appendix C just rewrites [Iyengar 05] proof. Maybe replace it by the proof for d-rectangular robust Bellman equation?
2. The robust partial concentrability dependence on the robust set $\Phi$: Is it tight? Is there a lower bound for this setting? The reason for this question is coming from the fact that the robust solution is looking for $\min$ over the robust set $\Phi$.
3. Honestly, Section B.2 can be expanded further. I found the following statement vague in its current form:
> our algorithm framework is unable to deal with this kind of rectangular robust sets in the context of partial coverage data due to some technical problems in applying the partial coverage coefficient (Assumption 3.3) under this kind of robust sets.

**Questions:**

please see weaknesses

**Limitations:**

Regarding S-rectangular sets, please see weaknesses

---

> ### Author Rebuttal · Authors · 2023-08-08
>
> **Response to Reviewer F7Eb**
>
> Thanks so much for your appreciation of our work! We will keep improving our paper following your suggestions. In the following, we address all your concerns and questions.
>
>
> **Q1: The robust Bellman equation for $d$-rectangular sets are not formally proven. The proof isn't there even in [1] to the extent of my search. The current Appendix C just rewrites [2] proof. Maybe replace it by the proof for d-rectangular robust Bellman equation?**
>
> **A1:** Yes, we agree with your comments that currently a strict proof of the robust Bellman equation for $d$-rectangular linear MDPs proposed by [1] is still missing. Actually we can prove it following a similar argument as in $\mathcal{S}\times\mathcal{A}$-robust set case. Thanks for your suggestions and we would consider adding its proof in the revision to be self-content.
>
> **Q2: regarding the robust partial concentrability dependence on the robust set $\boldsymbol{\Phi}$, is it tight? Is there a lower bound for this setting? The reason for this question is coming from the fact that the robust solution is looking for min over the robust set $\boldsymbol{\Phi}$.**
>
> **A2:** We think the robust concentrability dependence on the robust set is tight. For the robust single-policy clipped concentrability $C_{\mathrm{rob}}^\star$ defined in [3], we can show that $\sqrt{C_{P^\star, \boldsymbol{\Phi}}^\star} \le C_{\mathrm{rob}}^\star.$ Together with the lower bound $\Omega(C_{\mathrm{rob}}^\star/\varepsilon^2)$ in [3], we know our robust partial coverage coefficient $C_{P^\star, \boldsymbol{\Phi}}^\star$ characterizes the statistical limit of robust offline RL. Here we omit other parameters such as the horizon length $H$ in the lower bound and $\varepsilon$ is the accuracy of the desired policy.
>
> **Q3: Honestly, Section B.2 can be expanded further. I found the following statement vague in its current form: "our algorithm framework is unable to deal with this kind of rectangular robust sets in the context of partial coverage data due to some technical problems in applying the partial coverage coefficient (Assumption 3.3) under this kind of robust sets."**
>
> **A3:** Thanks for your suggestion! We will make the statement clearer in our revision. In the following we briefly explain the technical problems we met. Intuitively, for $\mathcal{S}$-rectangular RMDPs, it actually obeys another form of robust Bellman equation (RBE):
>
> $$
>    V_{h,P,\boldsymbol{\Phi}}^{\pi}(s_h) = \mathbb{E}_{a_h\sim \pi_h(\cdot|s_h)}[R_h(s_h,a_h)] + \inf\_{\widetilde{P}_h\in\boldsymbol{\Phi} (P_h)} \mathbb{E}\_{a_h\sim \pi_h(\cdot|s_h), s'\sim \widetilde{P}_h(\cdot|s_h,a_h)}[V\_{h+1,P,\boldsymbol{\Phi}}^{\pi}(s')]
> $$
>
> ($\mathcal{S}\times\mathcal{A}$-rectangular RMDPs also satisfy this form of RBE, but $\mathcal{S}$-rectangular RMDPs only satisfy this form of RBE). However, this form of RBE would not give the same suboptimality decomposition as we did by our proof techniques (Eqn. (D.8) in Appendix D), which is key to apply the robust partial coverage condition adopted by our paper (Assumption 3.3). Therefore, currently we are not sure of whether or not it is actually possible to include $\mathcal{S}$-rectangular RMDPs to our theoretical framework. It is an interesting future work for us to figure this out.
>
> **References:**
>
> [1] Ma, Xiaoteng, et al. "Distributionally robust offline reinforcement learning with linear function approximation." arXiv preprint arXiv:2209.06620 (2022).
>
> [2] Iyengar, Garud N. "Robust dynamic programming." Mathematics of Operations Research 30.2 (2005): 257-280.
>
> [3] Laixi Shi and Yuejie Chi. Distributionally robust model-based offline reinforcement learning with near-optimal sample complexity. arXiv preprint arXiv:2208.05767, 2022.

---

> > ### Comment · Reviewer_F7Eb · 2023-08-20
> >
> > The rebuttal addressed my concerns. I think my current rating considering the rebuttal and other reviewers’ concerns still stands correct, and I am positive for its publication.

---

> > > ### Author Response · Authors · 2023-08-20
> > >
> > > Thanks so much for your efforts reviewing our paper and your positive feedbacks! We will further improve our paper following your suggestions during revision.

---

### Official Review · Reviewer_otjP · 2023-06-30

**Soundness:** 4 excellent
**Presentation:** 4 excellent
**Contribution:** 3 good
**Rating:** 8
**Confidence:** 4

**Summary:**

This paper studies distributionally robust offline reinforcement learning. They propose a general learning principle, double pessimism, as well as a generic algorithm framework P$^2$MPO for robust offline RL, and  show that it is provably efficient in the context of general function approximation.

**Strengths:**

The paper is well-written and theoretically solid. It provides several novel contributions to the field of distributionally robust MDP.

First, it proposes a general learning principle, double pessimism, together with a generic algorithm framework P$^2$MPO and a unified theoretical analysis.

Second, it proposes several novel structures of uncertainty set and discusses their qualities under three commonly used rectangularity assumptions. Under these structures, they solve the open problem of learning robust offline RL in the context of general function approximation.


**Weaknesses:**

Some minor issue of typos.

**Questions:**

1. On line 100, is the policy a function from $\mathcal{S}$ to $\Delta(A)$?

2. How strong are assumption D.1 and E.1, are they reasonable?

3. Check inequality (E.23) '$0\leq \lambda_iH$'.


**Limitations:**

Yes.

---

> ### Author Rebuttal · Authors · 2023-08-08
>
> **Response to Reviewer otjP**
>
> Thanks you so much for your appreciation of our work! We will keep improving our paper following your feedbacks. In the following, we address all your concerns and questions.
>
> **Q1: Some minor typos: i) On line 100, is the policy a function from $\mathcal{S}$ to $\Delta(\mathcal{A})$? ii) Check inequality (E.23) $0\leq \lambda_i H$.**
>
> **A1:** Thanks so much for pointing those out! We will correct them in the revision.
>
> **Q2: How strong are assumption D.1 and E.1, are they reasonable?**
>
> **A2:** You mean the assumptions on the lower bound of dual variables (Assumption E.3 and F.2)? It's worth mentioning that the adoption of this assumption in problems with the KL robust set is consistent with previous works (e.g., [1]). From our perspective, removing this assumption in the context of RMDPs with function approximation, without introducing additional assumptions, presents a challenge due to the inherent nature of KL-divergence. In our work, we also investigate robust RL with the TV robust set, which is a standard and significant setting. For robust RL with TV robust set, we do NOT need the regular assumption like Assumption E.3 and F.2, and the final suboptimality gap is *polynomial in all parameters*.
>
> **References:**
>
> [1] Ma, Xiaoteng, et al. "Distributionally robust offline reinforcement learning with linear function approximation." arXiv preprint arXiv:2209.06620 (2022).

---

> > ### Comment · Reviewer_otjP · 2023-08-21
> >
> > The authors have addressed my questions and I will keep my score.

---

### Official Review · Reviewer_p1MD · 2023-07-03

**Soundness:** 3 good
**Presentation:** 2 fair
**Contribution:** 2 fair
**Rating:** 6
**Confidence:** 4

**Summary:**

This paper proposed a generic framework to study distributionally robust offline reinforcement learning problems, which included a model estimation step and a robust policy optimization step. Previous works in the literature usually assume finite state action spaces; this framework can incorporate function approximations and ultimately paved the way of tackling large state action spaces. Additionally, two specific model estimation approaches are introduced and studied in details; corresponding sample complexity results are provided.

**Strengths:**

This paper explicitly considered the two sources of uncertainties in distirbutionally robust RL problems. The first one is the difference between the estimated training model using finite samples and the true training model; the second one is the difference between the true training model and the testing model. Current literature usually assume that the testing model is within a radius of the estimated training model. The double pessimistic idea is new. Additionally, only partial coverage offline dataset is required, while many previous works require observing all state action pairs in the dataset.

This paper also provided an interesting sub-optimality theorem that decomposes the sub-optimality gap into a model estimation error part and a data coverage part, which could be used for future developments.

The problem setting, definitions and notations are clear and easy to understand.

**Weaknesses:**

The paper is somewhat hard to follow. A lot of examples that can be studied under the proposed framework are presented in the main paper. I think it is better to present one example in details in the main paper, move the others to the appendix and put relevant literature review in the main paper. Similarly, in the model estimation parts, one example should be enough.

The motivation of considering the double pessimism is not clear to me. Two uncertainty sets are introduced, the model uncertainty set, for example, is controlled by $\epsilon$ when using MLE estimator and the distribution shift robust uncertainty set is controlled by $\rho$. It is not clear to me whether the two phase uncertainty sets are necessary as in general we don't know $\rho$. In practice, people may use similar datasets that include both training and testing sets to estimate (guess) the uncertainty set radius, which is a single phase approach and directly captures the difference between the finite training set and the testing set. More rationales or examples should be provided to motivate the two-step approach. For example, 1) the authors could provide applications that singles-step uncertainty set cannot be easily estimated while the two-step approach is more applicable in practice, 2) the author could theoretically show that this two-step approach can avoid conservativeness under certain conditions.

**Questions:**

1.  The underscore $\lambda$ and $C_1$ in the Corollary 4.1 is not defined in the main paper. They are well-defined in the appendix; I think it is better to at least provided the meaning of underscore $\lambda$ here. In addition, is it possible to avoid the assumption of the lower bound on the dual variable $\lambda$? As you consider the two-step uncertainty set, I think you could provide sub-optimality gap in terms of $\epsilon$ and $\rho$ in Section 4 and discuss the choices of them as well.

2. Some distributionally robust RL papers that adopt KL-divengence as the uncertainty set measure have radius square in the SubOpt term, e.g., [37] [64] in the papers you cited, while your SubOpt does not suffer from the radius square term. Can you comment on this? I think this could be a big benefit when selecting small $\rho$s.

**Limitations:**

This is a theoretical work at this stage and thus no potential negative societal impact of their work.

---

> ### Author Rebuttal · Authors · 2023-08-08
>
> Thanks for your detailed review and the meaningful suggestions! In the following, we will try our best to address all your concerns and questions.
>
> **Q1: It is better to present one example in details in the main paper, move the others to the appendix and put relevant literature review in the main paper. Similarly for the model estimation parts.**
>
> **A1:** Thanks for pointing out! We will improve the presentation following your suggestions.
>
> **Q2: The motivation of double pessimism is not clear. It's unknown whether the two phase uncertainty sets are necessary as in general we don't know $\rho$.  ···**
>
> **A2:** Thanks for your questions. In the following, we will explain more about the motivation and necessity of the double pessimism principle.
>
> The theoretical motivation to consider performing pessimism twice is that we want to handle the two sources of distributional shifts in robust offline RL: (i) the mismatch between the behavior policy and the target policies to be learned; (ii) the mismatch between the training environment and the testing environment. Thus our approach is to perform pessimism in the face of *model estimation uncertainty* and *test environment uncertainty* simultaneously. The model estimation uncertainty originates from statistical estimation of the training environment transition kernel $P^{\star}$ under the mismatch between the state-action distributions induced by the behavior policy and the target policies. The test environment uncertainty comes from the the mismatch between the environments for training and testing. Since the test environemt is in a robust set centered at the training environment $P^{\star}$, these two kinds of uncertainties are coupled with each other and a two-step-style pessimism approach is developed.
>
> Theoretically, our double pessimism approach can reduce the requirement of the training data to the minimum: the robust partial coverage condition (Assumption 3.3), i.e., only covering the distributions induced by the optimal policy and the transition kernels in the robust set of the nominal transition kernel, which is much weaker than full-coverage-style conditions like a uniformly lower bounded data distribution. This is impossible without the double pessimism approach.
>
> Regarding your concern that $\rho$ might be unknown, we admit that in practice this could be the case. In that case, the robust parameter can either serve as a tuning parameter balancing between the robustness of and the performance of the learned policy, or simply be chosen by experts or priors. Still, in our theoretical work we assume a known $\rho$, which is commonly adopted by the large body of researches on robust RL.
>
> Finally, we clarify that our work focuses on the offline setup with no access to the test environment data, which is often the case in practice. E.g., in robotics, people may have no access to the exact place where the robotics they trained will be deployed. Thus no test data are available. It's an interesting future work if the learner is provided with an extra data related to the test environment.
>
> **Q3: The underscore $\underline{\lambda}$ and $C_1$ in the Corollary 4.1 is not defined in the main paper. I think it is better to provide the meaning of underscore $\underline{λ}$ here. In addition, is it possible to avoid the assumption of the lower bound on the dual variable $\underline{\lambda}$?**
>
> **A3:** We have mentioned in Line 293 that $C_1$ is an absolute constant. Furthermore, we acknowledge the importance of clarifying the meaning of $\underline{\lambda}$ in Corollary 4.1. In the revision, we will provide an explanation that $\underline{\lambda}$ represents the lower bound of the dual variables of some DRO problems.
>
> We want to emphasize that this issue is incurred by the KL robust set. In addition to this setting, we also study robust RL with TV robust set, which is also a standard and important setting. For robust RL with TV robust set, we do NOT need the regular assumption like Assumption E.3, and the final suboptimality gap is *polynomial in all parameters*. It's worth mentioning that the adoption of this assumption in problems with the KL robust set is consistent with previous works (e.g., [45]). From our perspective, removing this assumption in the context of RMDPs with function approximation, without introducing additional assumptions, presents a challenge due to the inherent nature of KL-divergence.
>
>
> **Q4: As you consider the two-step uncertainty set, I think you could provide sub-optimality gap in terms of $\epsilon$ and $\rho$ in Section 4 and discuss the choices of them as well.**
>
> **A4:** We note that the conclusions in Section 4 are actually in terms of the $\epsilon$ and $\rho$ you mentioned. The parameter $\xi$ in Corollary 4.1 & 4.2 corresponds to $\epsilon$. Therefore, for KL-divergence robust sets, the suboptimality gap is of order $\mathcal{O}(\xi\cdot\rho^{-1})$, while for the TV-distance robust sets the suboptimality gap is of order $\mathcal{O}(\xi)$, i.e., $\rho$-independent. The choice of $\xi$ is obtained from statistical analysis of the estimation of the nominal transition kernel, while $\rho$ is fixed by the RMDP problem instance we are considering. Thanks for pointing that out and we will add discussions.
>
> **Q5: Some DRRL papers with KL-divengence have radius square in the SubOpt term, e.g., [37] [64] you cited, while your SubOpt does not suffer from the radius square term. Can you comment on this?**
>
> **A5:** It seems that both [37] and [64] have Suboptimality scaling with $\mathcal{O}(\rho^{-1})$. Their results are presented in the form of sample complexity $N_{\mathrm{KL}} = \mathcal{O}(\epsilon^{-2}\rho^{-2})$, which means that to obtain $\epsilon$-optimal robust polciy, $N_{\mathrm{KL}}$ samples are needed. Converting it to the language of SubOpt, this gives an $\mathcal{O}(\rho^{-1}\cdot N_{\mathrm{KL}}^{-1/2})$ suboptimality. Thus, in the tabular case, our dependence on $\rho$ conincides with [37] and [64].

---

> > ### Comment · Reviewer_p1MD · 2023-08-18
> >
> > The authors have addressed all my concerned and I raised the final score.

---

> > > ### Author Response · Authors · 2023-08-20
> > >
> > > Thank you very much for your valuable feedbacks and updating your score! We will keep improving our work following your suggestions in the revision.

---

### Official Review · Reviewer_NCKb · 2023-07-04

**Soundness:** 3 good
**Presentation:** 3 good
**Contribution:** 2 fair
**Rating:** 4
**Confidence:** 4

**Summary:**

This paper studies the offline robust RL problem. A double pessimism approach is proposed and studied.

**Strengths:**

1. The approach is novel and new compared to previous offline robust RL ones.
2. The approach can be used for large-scale problems.
3. The theoretical analysis is comprehensive.

**Weaknesses:**

1. The model the authors proposed, seems hard to solve. With no robustness consider, the model reduces to the one in [Masatoshi Uehara and Wen Sun, 2022]. The model in the non-robust is hard to solve and becomes hard together with robustness.
2. Compared to [Masatoshi Uehara and Wen Sun, 2022], the contribution seems a little bit incremental to me. The results and approaches are not surprising, in aspects of approach designing and error bound analysis.

**Questions:**

1. Do you have some efficient approach to solve the model you proposed?
2. The analysis seems similar to the ones in [Masatoshi Uehara and Wen Sun, 2022], can you highlight the novelty and contribution?

**Limitations:**

See parts above.

---

> ### Author Rebuttal · Authors · 2023-08-08
>
> **Response to Reviewer NCKb**
>
> Thanks for your review and the feedback. We will try our best to address all your concerns and questions in the following.
>
> **Q1: The model the authors proposed, seems hard to solve. Do you have some efficient approach to solve the model you proposed?**
>
> **A1:** We clarify that our work focuses on the statistical side of robust offline RL under general function approximations, which remains an important open problem. Our algorithm is therefore information-theoretic and is indeed intractable with an abstract function approximation class. The objective we designed is to achieve statistical efficiency in the most general setup. Computational efficiency is not our focus.
>
> Meanwhile, if we specify the model space, for example, tabular case or linear/kernel function classes, we can design approximations of our doubly pessimistic algorithm. For example in tabular RMDPs, we can replace the iterative infimum in our double pessimism objective by an LCB-style bonus. This has been shown to be efficient both theoretically and experimentally by [1], which can demonstrate the efficiency and practicality of our general algorithm design. For more complicated deep RL setup, a promising approach is to extend the algorithm proposed by [2]. It implements the model-based pessimistic offline RL algorithm for non-robust MDPs, which iterates between an agent update step (corresponding to $\sup_{\pi\in\Pi}$) and an adversarial model update step (corresponding to $\inf_{P\in\widehat{\mathcal{P}}}$). Adapting the adversarial model update step to our doubly pessimistic value estimator can serve as an approximate implementation of the algorithm proposed by our work.
>
> **Q2: Compared to [3], the contribution in terms of approach designing and error bound analysis seems a little bit incremental. Can you highlight the novelty and contribution?**
>
> **A2:** We respectfully disagree that our work contributes incrementally compared to [3]. Essentially, our work is on distributionally robust offline reinforcement learning, a different problem setup from [3], with its own distinct challenges in terms of algorithmic design and theoretical analysis. And to the best of our knowledge, our work is the first to propose a provably sample-efficient algorithm for distributionally robust offline RL in the context of general function approximation.
>
> In the following, we compare our work with [3] in more detail.
>
> - **Approach designing.** In robust offline RL, there exist two sources of distributional shifts which are coupled with each other: (i) the mismatch between the behavior policy and the target policies to be learned; and (ii) the mismatch between the nominal environment and the perturbed environment. The latter is a unique challenge that is not presented in non-robust offline RL [3]. Therefore, it remains unknown how to design sample-efficient algorithms that can provably tackle these two types of shifts under general function approximation. With this in mind, our approach features a noval algorithmic design principle named "double pessimism", which performs pessimistic model selection in the face of both kinds of distributional shifts *simultaneously*. This is essentially different from [3]. Our work is the first to identify such a new algorithmic design principle for robust offline RL with general function approximation.
> - **Error bound analysis.** When there are coupled shifts (i) and (ii), the theoretical analysis of [3] would also fail. In fact, our analysis is based on a different framework from [3], in terms of analyzing:
>     - *Pessimism in the face of two souces of distributional shifts*: These calls for new analysis techniques for error decomposition and analysis of pessimism (Appendix D). Also, our analysis is based on the notion of robust partial coverage coefficient (Assumption 3.3) which is customized for robust RL. This is different from using the standard partial coverage coefficient for MDPs [3], requiring new analysis techniques.
>     - *Model estimation error analysis coupled with distributional shifts:* Compared with standard offline RL analysis of transition kernel estimation [3], this requires a delicate application and analysis of the dual representations for distributionally robust objectives (Appendices E and F). Under our unified analysis framework, we customize different model estimation subroutines and their corresponding analysis for different kinds of RMDPs. Also, we highlight that our work studies several new examples of RMDPs, e.g., factored RMDPs, and their model estimation analysis are completely new.
>
>
> **References:**
>
> [1] Shi, Laixi, and Yuejie Chi. "Distributionally robust model-based offline reinforcement learning with near-optimal sample complexity." arXiv preprint arXiv:2208.05767 (2022).
>
> [2] Rigter, Marc, Bruno Lacerda, and Nick Hawes. "Rambo-rl: Robust adversarial model-based offline reinforcement learning." Advances in neural information processing systems 35 (2022): 16082-16097.
>
> [3] Uehara, Masatoshi, and Wen Sun. "Pessimistic Model-based Offline Reinforcement Learning under Partial Coverage." International Conference on Learning Representations. 2022.

---

> > ### Comment · Reviewer_NCKb · 2023-08-21
> >
> > I am acknowledging I have read your arguments and maintain my rating.

---

> ### Comment · Area_Chair_67mr · 2023-08-21
>
> Dear Reviewer,
>
> Please reply to the authors' rebuttal and ask any clarifying questions if you need.
>
> Thanks,
>
> Your AC

---

### Decision · Program_Chairs · 2023-09-21

**Decision:**

Accept (poster)

**Comment:**

This paper explores the realm of distributionally robust offline reinforcement learning by introducing a concept termed 'double pessimism', and a generic algorithmic framework. Notably, the paper is well-articulated, with a clear presentation and substantial theoretical contributions.

Following in-depth discussion among reviewers and authors, the majority have recommended acceptance of this paper, a view with which I agree. The authors are encouraged to integrate all points raised during the rebuttal phase into the final version of their manuscript.

Lastly, congratulations on this well-accomplished piece of work!